# An important role for triglyceride in regulating spermatogenesis

Charlotte F Chao[1†], Yanina-Yasmin Pesch[1†], Huaxu Yu[2], Chenjingyi Wang[2], Maria J Aristizabal[3], Tao Huan[2], Guy Tanentzapf[1], Elizabeth Rideout[1]*

[1]Department of Cellular and Physiological Sciences, Life Sciences Institute, The University of British Columbia, Vancouver, Canada; [2]Department of Chemistry, The University of British Columbia, Vancouver, Canada; [3]Department of Biology, Queen's University, Kingston, Canada

*For correspondence:
elizabeth.rideout@ubc.ca

[†]These authors contributed equally to this work

Competing interest: The authors declare that no competing interests exist.

**Abstract** *Drosophila* is a powerful model to study how lipids affect spermatogenesis. Yet, the contribution of neutral lipids, a major lipid group which resides in organelles called lipid droplets (LD), to sperm development is largely unknown. Emerging evidence suggests LD are present in the testis and that loss of neutral lipid- and LD-associated genes causes subfertility; however, key regulators of testis neutral lipids and LD remain unclear. Here, we show LD are present in early-stage somatic and germline cells within the *Drosophila* testis. We identified a role for triglyceride lipase *brummer* (*bmm*) in regulating testis LD, and found that whole-body loss of *bmm* leads to defects in sperm development. Importantly, these represent cell-autonomous roles for *bmm* in regulating testis LD and spermatogenesis. Because lipidomic analysis of *bmm* mutants revealed excess triglyceride accumulation, and spermatogenic defects in *bmm* mutants were rescued by genetically blocking triglyceride synthesis, our data suggest that *bmm*-mediated regulation of triglyceride influences sperm development. This identifies triglyceride as an important neutral lipid that contributes to *Drosophila* sperm development, and reveals a key role for *bmm* in regulating testis triglyceride levels during spermatogenesis.

## eLife assessment

This **important** study identifies a role for triglycerides and lipid droplets in spermatogenesis, with data supporting relevance of this finding across phyla. The work shows with **convincing** data that a triglyceride lipase is required cell-autonomously for germline differentiation into meiotic stages and haploid spermatids and that an increase in triglycerides is detrimental to spermatogenesis. This paper would be of interest to developmental and cell biologists working on gametogenesis.

## Introduction

Lipids play an essential role in regulating spermatogenesis across animals (**Brill et al., 2016**; **Keber et al., 2013**; **Rato et al., 2012**; **Wang and Huang, 2012**). Studies in *Drosophila* have illuminated key roles for multiple lipid species in regulating sperm development (**de Cuevas and Matunis, 2011**; **Demarco et al., 2014**; **Fabian and Brill, 2012**). For example, phosphatidylinositol and its phosphorylated derivatives participate in diverse aspects of *Drosophila* spermatogenesis including meiotic cytokinesis (**Brill et al., 2016**; **Brill et al., 2000**; **Giansanti et al., 2006**; **Wong et al., 2005**; **Wong et al., 2007**), somatic cell differentiation (**Amoyel et al., 2016**), germline and somatic cell polarity maintenance (**Fairchild et al., 2017**; **Inaba et al., 2015**; **Krahn et al., 2010**; **Papagiannouli et al., 2019**), and germline stem cell (GSC) maintenance and proliferation (**Ueishi et al., 2009**). Membrane lipids also influence sperm development (**Phan et al., 2007**; **Steinhauer et al., 2009**), whereas fatty acids play a

role in processes such as meiotic cytokinesis (*Szafer-Glusman et al., 2008*) and sperm individualiza-tion (*Ben-David et al., 2015*; *Jung et al., 2007*). While these studies suggest key roles for membrane lipids and fatty acids during *Drosophila* spermatogenesis, some of which are conserved in mammals (*Rabionet et al., 2015*; *Santiago Valtierra et al., 2018*; *Zadravec et al., 2011*), much less is known about how neutral lipids contribute to spermatogenesis.

Neutral lipids are a major lipid group that includes triglyceride and cholesterol ester, where neutral lipids reside within specialized organelles called lipid droplets (LD) (*Walther and Farese, 2012*). LD are found in diverse cell types (*e.g.* adipocytes, muscle, liver, glia, and neurons) (*Wat et al., 2020*; *Thiele and Spandl, 2008*; *Walther and Farese, 2012*), and play key roles in maintaining cellular lipid homeostasis. In nongonadal cell types, correct regulation of LD contributes to cellular energy produc-tion (*Bosma, 2016*; *Grönke et al., 2007*; *Rambold et al., 2015*), sequestration and redistribution of lipid precursors (*Dichlberger et al., 2014*; *Mitsche et al., 2018*; *Rajakumari et al., 2010*; *Schlager et al., 2015*; *Zanghellini et al., 2008*), and regulation of lipid toxicity (*Bailey et al., 2015*; *Liu et al., 2017*; *Nguyen et al., 2017*). The importance of LD to normal cellular function in nongonadal cell types is shown by the fact that dysregulation of LD causes defects in cell differentiation, survival, and energy production (*Walther and Farese, 2012*; *Bailey et al., 2015*; *Henne, 2019*; *Welte, 2015*). In the testis, much less is known about the regulation and function of neutral lipids and LD, and how this regulation affects sperm development.

Multiple lines of evidence suggest a potential role for neutral lipids and LD during spermatogen-esis. First, genes that encode proteins associated with neutral lipid metabolism and LD are expressed in the testis across multiple species (*Casado et al., 2013*; *El-Shehawi et al., 2020*; *Wang et al., 2015*). Second, testis LD have been identified in mammals and flies under both normal physiological conditions (*Wat et al., 2020*; *Wang et al., 2015*; *Bajpai et al., 1998*; *Kerr and De Kretser, 1975*; *Mori and Christensen, 1980*; *Paniagua et al., 1987*) and after mitochondrial stress (*Sênos Demarco et al., 2019*). Third, loss of genes associated with neutral lipid metabolism and LD cause subfertility phenotypes in both flies and mammals (*Wat et al., 2020*; *Chen et al., 2014a*; *Hermo et al., 2008*; *Masaki et al., 2017*). While studies suggest that mammalian testis LD contribute to steroidogenesis (*Wang et al., 2017*), the spatial, temporal, and cell-type-specific requirements for neutral lipids and LD in the testis have not been explored in detail in any animal. It remains similarly unclear which genes are responsible for regulating neutral lipids and LD during spermatogenesis.

To address these knowledge gaps, we used *Drosophila* to investigate the regulation and function of neutral lipids and LD during sperm development. Our detailed analysis of spermatogenesis under normal physiological conditions revealed the presence of LD in early-stage somatic and germline cells in the testis. We identified triglyceride lipase *brummer* (*bmm*) as a regulator of testis LD, and showed that this represents a cell-autonomous role for *bmm* in the germline. Importantly, we found that *bmm*-mediated regulation of testis LD was significant for spermatogenesis, as both whole-body and cell-autonomous loss of *bmm* caused defects in sperm development. Given that our lipidomic analysis revealed an excess accumulation of triglyceride in animals lacking *bmm*, and that genetically blocking triglyceride synthesis rescued many spermatogenic defects associated with *bmm* loss, our data suggest that *bmm*-mediated regulation of triglyceride is important for normal *Drosophila* sperm development. This reveals previously unrecognized roles for neutral lipids such as triglyceride in regu-lating spermatogenesis, and for *bmm* in regulating sperm development under normal physiological conditions. Together, these findings advance knowledge of the regulation and function of neutral lipids during spermatogenesis.

## Results

### LD are present in early-stage somatic and germline cells

We previously reported the presence of small LD (<1 µm) at the apical tip of the testis (*Wat et al., 2020*), a finding we reproduced in *w^1118* males using neutral lipid stain BODIPY (*Figure 1A*). These LD were present in the region that contains stem cells, early-stage somatic cells, and germline cells (*Figure 1A, A'*, arrows). LD were also present in the hub, an organizing center and stem cell niche in the *Drosophila* testis (*Figure 1A'', A'''*, arrows) (*de Cuevas and Matunis, 2011*), but largely absent within the area occupied by spermatocytes (*Figure 1A, A'*, arrowheads). This LD distribution was reproduced in two independent genetic backgrounds and at two additional ages (*Figure 1B, C*). While

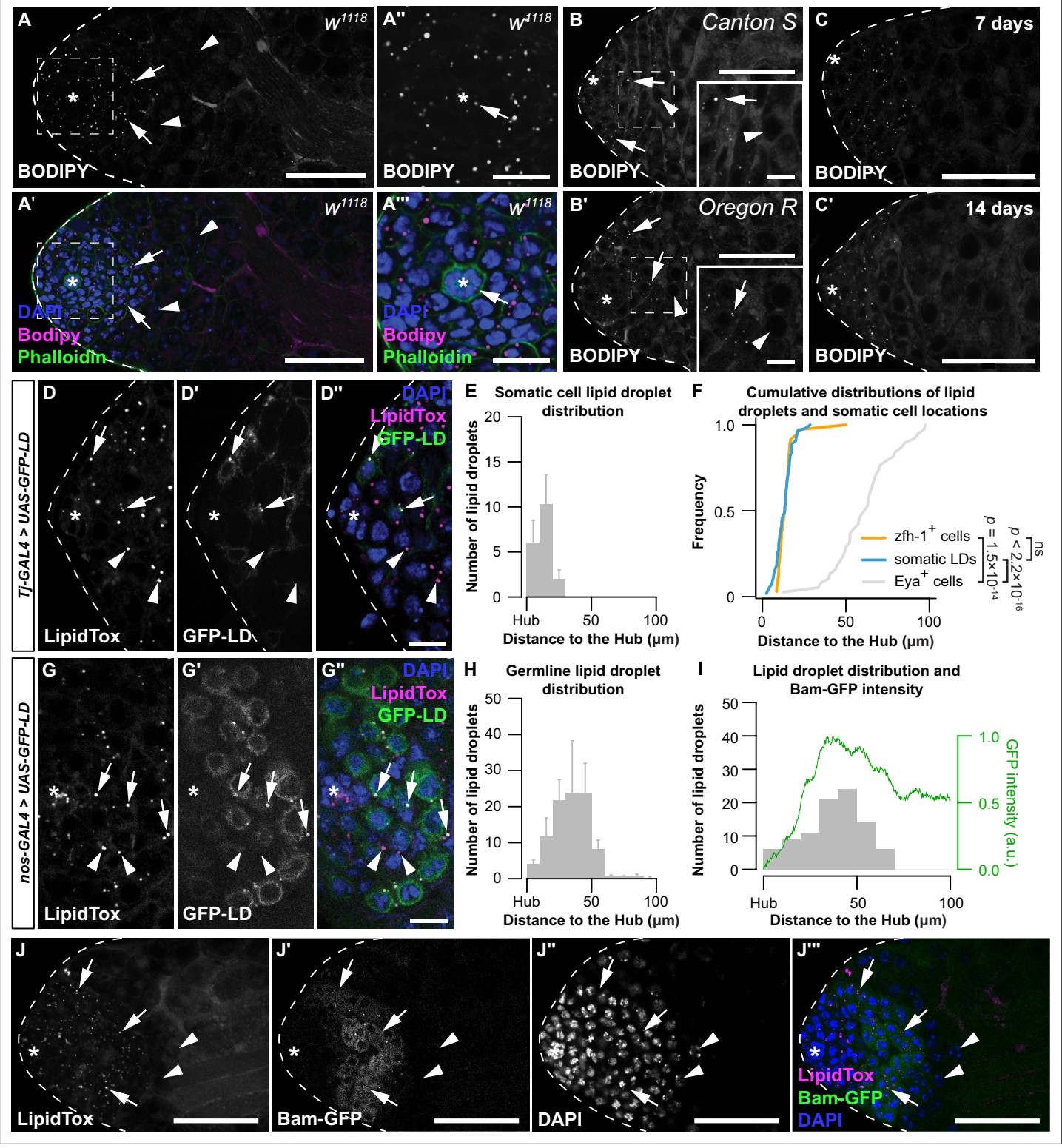

**Figure 1.** Lipid droplets (LD) are present in early-stage somatic and germ cells. (**A**) Testis LD in $w^{1118}$ animals visualized with neutral lipid dye BODIPY. (**A, A'**) Scale bar = 50 μm; (**A'', A'''**) scale bar = 15 μm. Asterisk indicates hub in all images. Arrows point to LD; arrowheads point to spermatocytes in A, B. Spermatocytes were identified as described in methods section. (**B**) Testis LD visualized with BODIPY in newly eclosed males from two wild-type genotypes. Scale bars: main image = 50 μm; inset image = 10 μm. (**C**) Testis LD from $w^{1118}$ animals at different times post-eclosion. Scale bars = 50 μm. (**D**) Testis LD visualized with LipidTox Red in animals with somatic cell overexpression of GFP-LD (*Tj-GAL4>UAS-GFP-LD*). Green fluorescent protein (GFP)- and LipidTox Red-positive punctae are somatic LD (D–D'' arrows); LipidTox punctae without GFP indicate germline LD (D–D'' arrowheads). Scale

*Figure 1 continued on next page*

*Figure 1 continued*

bars = 10 μm. (**E**) Histogram showing the spatial distribution of somatic cell LD; error bars represent standard error of the mean (SEM). (**F**) Cumulative frequency distributions of somatic LD (blue line, data reproduced from E), zfh-1-positive somatic cells (zfh-1+ cells, orange line), and Eya-positive somatic cells (Eya+ cells, gray line). (**G**) Testis LD visualized with LipidTox Red in males with germline overexpression of GFP-LD (*nos-GAL4>UAS-GFP-LD*). GFP- and LipidTox Red-positive punctae indicate germline LD (arrows); LipidTox punctae without GFP indicate non-germline LD (arrowheads). Scale bars = 10 μm. (**H**) Histogram representing the spatial distribution of LD within the germline; error bars represent SEM. (**I**) Histogram representing the spatial distribution of LD and GFP fluorescence (green line) (arbitrary units, a.u.) in a representative testis of a *bam-GFP* animal (panel J). (**J**) Testis LD in a *bam-GFP* animal; arrows point to LD and arrowheads point to spermatocytes. Scale bar = 50 μm. See also *Figure 1—figure supplement 1*.

The online version of this article includes the following figure supplement(s) for figure 1:

**Figure supplement 1.** Cholesterol is absent from testis lipid droplets.

LD may contain multiple neutral lipid species (*Walther et al., 2017*), cholesterol-binding fluorescent polyene antibiotic filipin III did not detect cholesterol within testis LD (*Figure 1—figure supplement 1A*), suggesting triglyceride is the main neutral lipid in *Drosophila* testis LD.

*Drosophila* spermatogenesis requires the co-development and differentiation of the germline and the somatic lineages (*Boyle and DiNardo, 1995*). To identify LD in each lineage, we used the GAL4/UAS system to overexpress a GFP transgene fused to the LD-targeting motif of motor protein *Klarsicht* (*UAS-GFP-LD*) (*Yu et al., 2011*). Somatic overexpression of *UAS-GFP-LD* using *Traffic jam (Tj)-GAL4* revealed that the majority of somatic LD in 0-day-old males were located <30 μm from the hub (*Figure 1D, E*). Because the somatic LD distribution coincided with a marker for somatic stem cells and their immediate daughter cells (Zinc finger homeodomain 1, Zfh-1) (*Figure 1F*; two-sample Kolmogorov–Smirnov test) (*Leatherman and Dinardo, 2008*), but not with a marker for late somatic cells (Eyes absent, Eya) (*Amoyel et al., 2016*; *Fabrizio et al., 2003*), our data suggest LD are present in early somatic cells. Germline overexpression of *UAS-GFP-LD* using *nanos (nos)-GAL4* demonstrated the presence of LD within germline cells near the apical tip of the testis in 0-day-old males (*Figure 1G, H*). Specifically, the disappearance of germline LD coincided with peak expression of a GFP reporter that reflects the expression of Bag-of-marbles (Bam) protein in the testis (Bam-GFP) (*Chen and McKearin, 2003*; *Figure 1I, J*). Because peak Bam expression signals the last round of transient amplifying mitotic cell cycle prior to the germline's transition into the meiotic cell cycle (*Insco et al., 2009*; *Chen et al., 2014b*; *Gönczy et al., 1997*), our data suggest that germline LD, like somatic LD, are present in cells at early stages of development.

## *brummer* plays a cell-autonomous role in regulating testis LD

*Adipose triglyceride lipase* (*ATGL*) is a critical regulator of neutral lipid metabolism and LD (*Athenstaedt and Daum, 2003*; *Chitraju et al., 2013*; *Grönke et al., 2005*; *Haemmerle et al., 2006*; *Haemmerle et al., 2011*; *Huijsman et al., 2009*; *Korbelius et al., 2019*; *Kurat et al., 2006*; *Zimmermann et al., 2004*; *Attané et al., 2016*). Loss of *ATGL* in many cell types triggers LD accumulation, and *ATGL* overexpression decreases LD number (*Grönke et al., 2007*; *Grönke et al., 2005*; *Haemmerle et al., 2006*; *Korbelius et al., 2019*; *Zimmermann et al., 2004*; *Lee et al., 2014*; *Tuohetahuntila et al., 2016*). Given that the *Drosophila ATGL* homolog *brummer* (*bmm*) regulates testis LD induced by mitochondrial stress (*Sênos Demarco et al., 2019*), we explored whether *bmm* regulates testis LD under normal physiological conditions. We first examined *bmm* expression in the testis by isolating this organ from flies carrying a *bmm* promoter-driven *GFP* transgene (*bmm-GFP*) that recapitulates many aspects of *bmm* mRNA regulation (*Men et al., 2016*). GFP expression was present in the germline of *bmm-GFP* testes, and we found germline GFP levels were higher in spermatocytes than at earlier stages of sperm development (*Figure 2A, B*; one-way analysis of variance [ANOVA] with Tukey multiple comparison test). In further support of this observation, we analyzed a publicly available single-cell RNA sequencing dataset from the male reproductive organ (*Li et al., 2021*). Using pseudotime analysis, we arranged the germline (*Figure 2—figure supplement 1A*) and somatic cells (*Figure 2—figure supplement 1B*) based on their annotated developmental trajectory. The expression pattern of *bmm* in the germline matched our observation with the *bmm-GFP* reporter (*Figure 2—figure supplement 1C*). While levels of the *bmm-GFP* reporter were lower in somatic cells, single-cell RNA sequencing data identified *bmm* expression in the somatic lineage that was higher in cells at later stages of development (*Figure 2—figure supplement 1D*). Additional neutral lipid- and lipid droplet-associated genes such as *lipid storage droplet-2*, *Seipin*, *Lipin*, and *midway* also showed differential

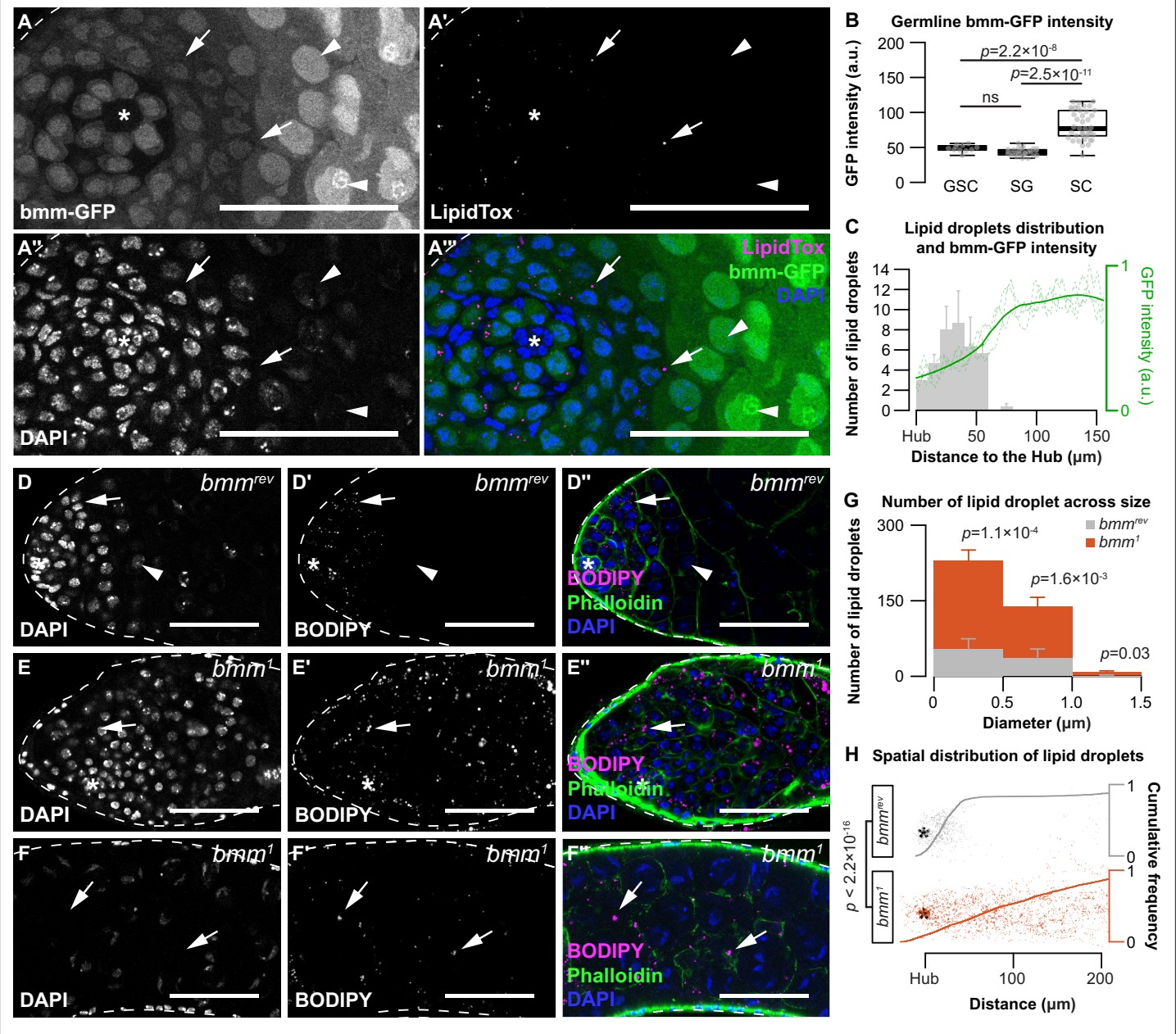

**Figure 2.** *brummer* regulates lipid droplets (LD) in both germline and somatic cells of the testis. (**A–A''''**) Testis LD indicated by LipidTox Red in *bmm-GFP* animals. Arrows point to LD in all images. Arrowheads point to spermatocytes. Scale bars = 50 µm. Asterisks indicate the hub in all images. (**B**) Quantification of nuclear GFP intensity in testes isolated from *bmm-GFP* animals (*n* = 3). Germline stem cell (GSC), spermatogonia (SG), spermatocyte (SC). (**C**) Spatial distribution of LD (gray histogram) and GFP expression (green line) in testes from *bmm-GFP* animals as a function of distance from the hub (*n* = 3). LD near the apical region of the testis in *bmm*^*rev*^ (**D**) or *bmm*^*1*^ (**E**) animals. (**F**) LD further away from the apical tip in *bmm*^*1*^ animals. (**D–F**) Scale bars = 50 µm. (**G**) Histogram representing testis LD size distribution in *bmm*^*rev*^ (gray) and *bmm*^*1*^ (orange). (B,C,G) Error bars represent standard error of the mean (SEM). (**H**) Apical tip of the testes is at the left of the graph; individual dots represent a single LD and its relative position to the hub marked by an asterisk. Cumulative frequency distribution of the distance between LD and the apical tip of the testes are drawn as solid lines. See also *Figure 2—figure supplement 1* and *Figure 2—figure supplement 2*.

The online version of this article includes the following figure supplement(s) for figure 2:

**Figure supplement 1.** Expression of *brummer* and selected lipid metabolic genes during spermatogenesis in germline and somatic lineages.

**Figure supplement 2.** *brummer* regulates lipid droplets (LD) in both germline and somatic cells of the testis.

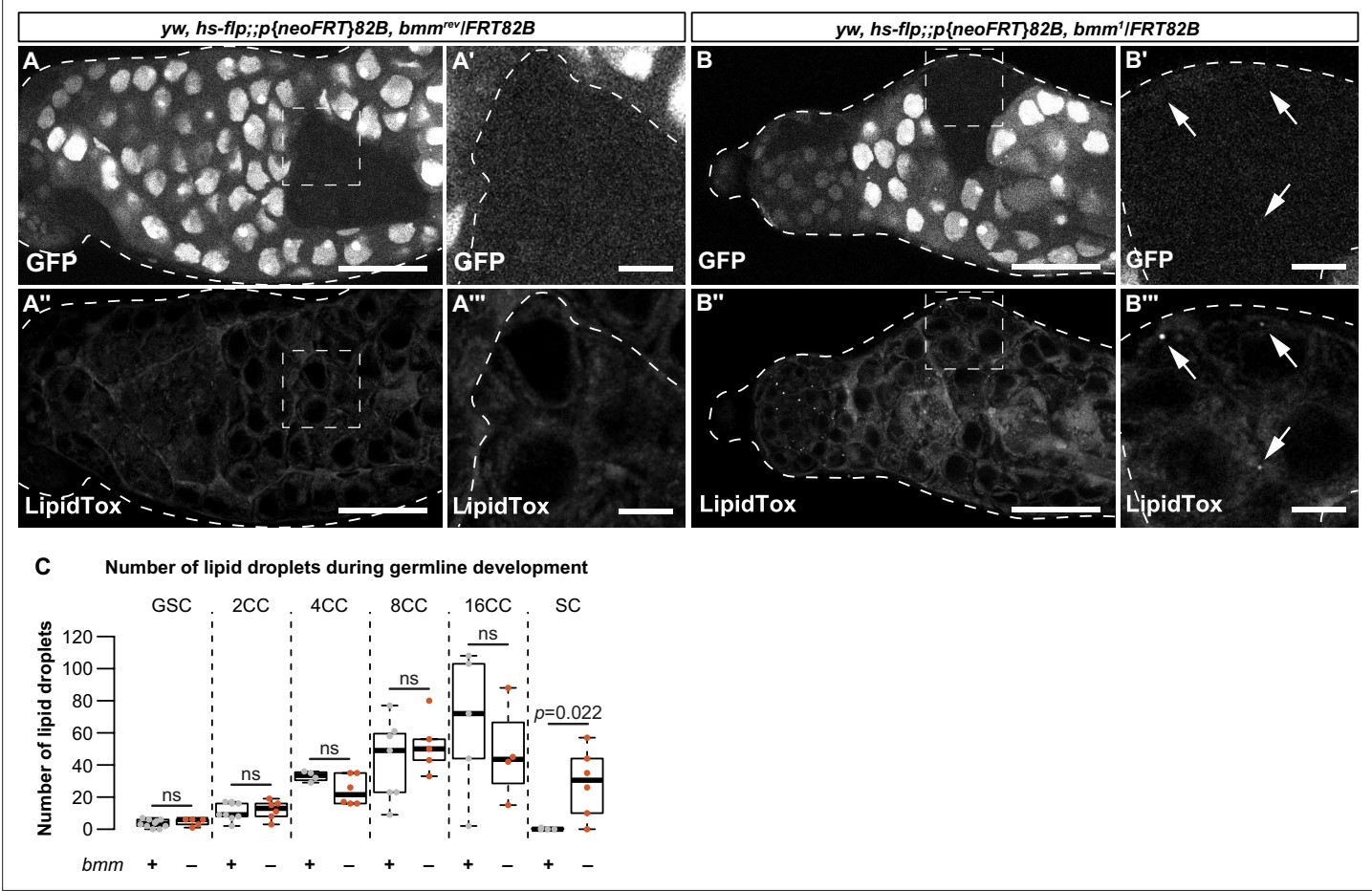

**Figure 3.** *bmm* regulates germline lipid droplets (LD) in a cell-autonomous manner. (**A, B**) Single confocal slices through a representative testis isolated from an individual carrying clones induced using the FLP-FRT system at 3 days post-clone induction. Clones are homozygous for an allele that encodes a functional *bmm* protein product (*bmm^rev^*; **A–A'''**) or a loss-of-function *bmm* allele (*bmm^1^*, **B–B'''**). GFP-negative areas mark homozygous clones in panels A and B; the boxed areas in A, A'' and B, B'' are shown in A', A''', and B', B''', respectively. In homozygous *bmm^rev^* spermatocyte clones we detected no LD using neutral lipid dye LipidTox (**A'', A'''**). In contrast, spermatocyte clones homozygous for *bmm^1^* have detectable LD (**B'', B'''**, arrows). Scale bars = 50 μm in A, A'' and B, B''; scale bars = 10 μm in A', A''' and B', B'''. (**C**) Number of testis LD in *bmm^rev^* (gray) or *bmm^1^* (orange) in FLP-FRT clones 3 days post-clone induction; dots represent measurements from a single clone. The number of cells in each cyst (CC) counted is indicated. There were significantly more LD in *bmm^1^* spermatocyte (SC) clones (p = 0.026; Welch two-sample *t*-test) but not at other stages of development. Error bars represent standard error of the mean (SEM).

regulation during differentiation in the testis (***Figure 2—figure supplement 1C, D***). Combined with our data on the location of testis LD, these gene expression data suggest that *bmm* upregulation in both somatic and germline cells during differentiation corresponds to the downregulation of testis LD. Supporting this, germline GFP levels were negatively correlated with testis LD in *bmm-GFP* flies (***Figure 2A, C***), suggesting regions with higher *bmm* expression had fewer LD.

To test whether *bmm* regulates testis LD, we compared LD in testes from 0-day-old males carrying a loss-of-function mutation in *bmm* (*bmm^1^*) to control male testes (*bmm^rev^*) (***Grönke et al., 2005***). *bmm^1^* males had significantly more LD across all LD sizes compared with control males at the apical tip of the testis (***Figure 2D–G***; Welch two-sample *t*-test with Bonferroni correction) and showed a significantly expanded LD distribution (***Figure 2D–F, H***; two-sample Kolmogorov–Smirnov test). This suggests *bmm* normally restricts LD to the apical tip of the testis, an observation we confirmed in both somatic and germline lineages using lineage-specific expression of GFP-LD (***Figure 2—figure supplement 2A–D***). Importantly, after inducing homozygous *bmm^rev^* or *bmm^1^* clones in the testes using the *FLP-FRT* system (***Figure 3A, B***; ***Xu and Rubin, 1993***), we found *bmm^1^* spermatocyte clones had significantly more LD at 3 days post-clone induction (***Figure 3C***; Welch two-sample *t*-test), a stage at which LD were absent from *bmm^rev^* clones. Because we observed no significant effect of cell-autonomous *bmm*

loss on LD at any other stage of germline development (*Figure 3C*), this suggests *bmm* function is not required to regulate LD at early stages of germ cell development. Instead, our data suggest *bmm* plays a role in regulating LD at the spermatogonia–spermatocyte transition. While we were unable to assess LD in *bmm¹* somatic clones, our data reveal a previously unrecognized cell-autonomous role for *bmm* as a regulator of LD in germline cells.

## *brummer* plays a cell-autonomous role in regulating germline development

To determine the physiological significance of *bmm*-mediated regulation of testis LD, we investigated testis and sperm development in males without *bmm* function. In 0-day-old *bmm¹* males reared at 25°C, testis size was significantly smaller than in age-matched *bmm^rev* controls (*Figure 4A, B*; Welch two-sample *t*-test), and the number of spermatid bundles was significantly lower (*Figure 4C*; Kruskal–Wallis rank sum test). When the animals were reared at 29°C, a temperature that exacerbates spermatogenesis defects associated with changes in lipid metabolism (*Ben-David et al., 2015*), *bmm¹* phenotypes were more pronounced (*Figure 4—figure supplement 1A, B*; Welch two-sample *t*-test, Kruskal–Wallis rank sum test). Defects in testis size were also observed at 14 days post-eclosion; suggesting testis size defects persist later into the life course (*Figure 4—figure supplement 1C*; Welch two-sample *t*-test). In contrast, the number of spermatid bundles per testis was not significantly different between *bmm¹* and *bmm^rev* males at this age (*Figure 4—figure supplement 1D*; Welch two-sample *t*-test), potentially due to a large decrease in the number of spermatid bundles in 14-day-old *bmm^rev* males (*Figure 4C*, *Figure 4—figure supplement 1D*). Together, these data suggest loss of *bmm* affects testis development and spermatogenesis. Similar phenotypes are observed in male mice without ATGL (*Masaki et al., 2017*), and supplementing the diet of *bmm¹* males with medium-chain triglycerides partially rescued the testis and spermatogenic defects we observed in flies (*Figure 4—figure supplement 1E, F*; one-way ANOVA with Tukey multiple comparison test), as it does in mice (*Masaki et al., 2017*; *Kim et al., 2017*). This identifies similarities between flies and mice in fertility-related phenotypes associated with whole-body loss of *bmm/ATGL*.

To explore spermatogenesis in *bmm¹* animals, we used an antibody against the germline cell-specific marker Vasa to visualize the germline in the testes of *bmm¹* and *bmm^rev* males (*Figure 4D, E*; *Lasko and Ashburner, 1988*). We observed a significant increase in the number of GSCs (*Figure 4F*; Kruskal–Wallis rank sum test) and higher variability in GSC number in *bmm¹* males (p = $5.7 \times 10^{-12}$ by *F*-test). Given that GSC number is affected by hub size and GSC proliferation (*Resende et al., 2013*; *Kiger et al., 2000*), we monitored both parameters in *bmm¹* and *bmm^rev* controls. While hub size in *bmm¹* testes was significantly larger than in testes from *bmm^rev* controls (*Figure 4—figure supplement 1G, H*; Welch two-sample *t*-test), the number of phosphohistone H3-positive GSCs, which indicates proliferating GSCs, was unchanged in *bmm¹* animals (*Figure 4—figure supplement 1I*; Kruskal–Wallis rank sum test). While this indicates a larger hub may partly explain *bmm*'s effect on GSC number, *bmm* also plays a cell-autonomous role in regulating GSCs, as we recovered a higher proportion of *bmm¹* clones in the GSC pool compared with *bmm^rev* clones at 14 days after clone induction (*Figure 4G*; Welch two-sample *t*-test). Given that we detected no effect of cell-autonomous *bmm* loss on the number of GSC LD (*Figure 3C*), more work will be needed to understand how *bmm* regulates GSCs at a stage prior to its effects on LD number. Future studies will also need to confirm whether *bmm¹* mutant GSCs show an increased ability to occupy space at the hub.

Beyond GSCs, we uncovered additional spermatogenesis defects in *bmm¹* testes. Peak Bam-GFP expression in germline cells of the testes from 0-day-old *bmm¹* and *bmm^rev* males showed GFP-positive cysts were significantly further away from the hub in *bmm¹* testes (*Figure 4H*, *Figure 4—figure supplement 1J*; Welch two-sample *t*-test). Indeed, 15/18 *bmm¹* testes contained Vasa-positive cysts with large nuclei in the distal half of the testis (*Figure 4I*, arrows), a phenotype not present in *bmm^rev* testes (0/8) (p = 0.0005 by Pearson's Chi-square test). Because these phenotypes are also seen in testes with differentiation defects (*Fairchild et al., 2017*; *Lin et al., 1996*), we recorded the stage of sperm development reached by the germline in *bmm¹* testes. Most *bmm¹* testes contained post-meiotic cells in males raised at 25°C (*Figure 4—figure supplement 1K*); however, germline development did not progress past the spermatocyte stage in most *bmm¹* testes from animals raised at 29°C (*Figure 4—figure supplement 1K*). Testes from *bmm¹* males reared at 25°C also had a smaller Boule-positive area (*Figure 4J*, *Figure 4—figure supplement 1L*; Welch two-sample *t*-test) and fewer

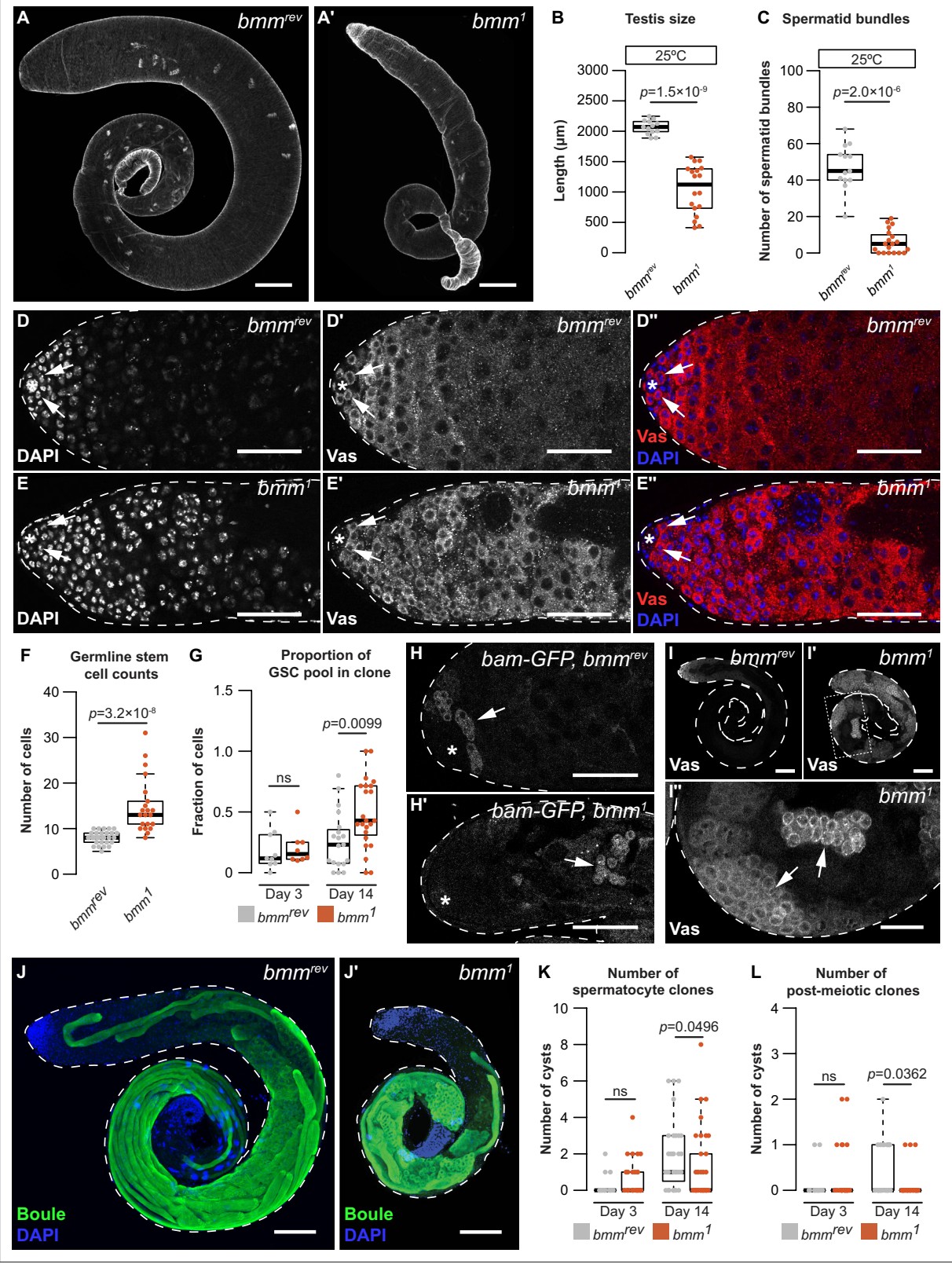

**Figure 4.** A cell-autonomous role for *bmm* in regulating spermatogenesis. Testes isolated from *bmm^rev^* (**A**) and *bmm^1^* (**A'**) animals raised at 25°C stained with phalloidin. Scale bars = 100 μm. (**B**) Testis size in *bmm^1^* and *bmm^rev^* animals raised at 25°C. (**C**) Spermatid bundle number in *bmm^1^* and *bmm^rev^* testes from animals reared at 25°C. Representative images of *bmm^rev^* (**D**) or *bmm^1^* (**E**) testes stained with 4',6-diamidino-2-phenylindole (DAPI) and anti-Vasa antibody. Arrows indicate germline stem cells (GSCs). Scale bar = 50 μm. The hub is marked by an asterisk in all images. (**F**) GSC number in *bmm^1^*

*Figure 4 continued*

and *bmm*[rev] testes. (**G**) Proportion of GSCs that were either *bmm*[1] or *bmm*[rev] clones at 3 and 14 days post-clone induction. (**H**) Representative images of *bmm*[rev] (**H**) and *bmm*[1] (**H'**) testes carrying *bam-GFP*; data quantified in **Figure 4—figure supplement 1J**. Arrows indicate regions with high Bam-GFP. Scale bars = 50 µm. (**I**) Representative images of *bmm*[rev] (**I**) or *bmm*[1] (**I', I''**) testes stained with anti-Vasa antibody. Arrows indicate Vasa-positive cysts in *bmm*[1] testis. Panel I'' is magnified from the boxed region in I'. (**I, I'**) Scale bars = 100 µm; (**I''**) scale bar = 50 µm. Maximum projection of *bmm*[rev] (**J**) or *bmm*[1] (**J'**) testes stained with anti-Boule antibody (green) and DAPI (blue). Scale bars = 100 µm. Number of *bmm*[1] and *bmm*[rev] spermatocyte clones (**K**) or post-meiotic clones (**L**) at 3 and 14 days post-clone induction. (B,C,F,G,K,L) Error bars indicate standard error of the mean (SEM). See also **Figure 4—figure supplement 1**.

The online version of this article includes the following figure supplement(s) for figure 4:

**Figure supplement 1.** Additional characterization of testis development and spermatogenesis in animals lacking *bmm*.

individualization complexes and waste bags (**Figure 4—figure supplement 1M, N**; Kruskal–Wallis rank sum test). Because Boule-positive area, individualization complexes, and waste bags are all markers for later stages in sperm development, these data indicate that loss of *bmm* caused a reduction in differentiated cell types. Because we observed significantly fewer *bmm*[1] spermatocyte and spermatid clones at 14 days after clone induction (**Figure 4K, L**; p = 0.0496, Kruskal–Wallis rank sum test), these effects on germline development may represent a cell-autonomous role for *bmm* in regulating spermatogenesis in this cell type. Given that the statistical significance of this finding was not as strong as for our other data, future studies should repeat this experiment with more samples. We also reveal a potential non-cell-autonomous role for somatic *bmm*. While there was no difference in the ratio of Zfh-1-positive cells between homozygous clones and heterozygous clones in animals carrying the *bmm*[1] or *bmm*[rev] alleles at 14 days post-clone induction (**Figure 4—figure supplement 1O**; Kruskal–Wallis rank sum test), the distance from the hub to the Zfh-1-positive clones was significantly decreased in *bmm*[1] homozygous clones (**Figure 4—figure supplement 1P**; Kruskal–Wallis rank sum test). Together, these data indicate *bmm* may play a cell-autonomous role in germline cells, and potentially a non-cell-autonomous role in somatic cells, to regulate spermatogenesis.

## *brummer*-dependent regulation of testis triglyceride levels affects spermatogenesis

*ATGL* catalyzes the first and rate-limiting step of triglyceride hydrolysis (**Zimmermann et al., 2004**; **Eichmann et al., 2012**; **Schweiger et al., 2006**). Loss of this enzyme or its homologs leads to excess triglyceride accumulation (**Wat et al., 2020**; **Grönke et al., 2007**; **Grönke et al., 2005**; **Zimmermann et al., 2004**; **Lee et al., 2014**) and shifts in multiple lipid classes (**Chitraju et al., 2013**; **Missaglia et al., 2017**; **Williams et al., 1991**; **Yang et al., 2020**). To determine how loss of *bmm* affects spermatogenesis, we carried out whole-body mass spectrometry (MS)-based untargeted lipidomic profiling of *bmm*[1] and *bmm*[rev] males. Hierarchical clustering of lipid species suggests that *bmm*[1] and *bmm*[rev] males show distinct lipidomic profiles (**Figure 5A**). Overall, we detected 2464 and 1144 lipid features with high quantitative confidence in positive and negative ion modes, respectively. By matching experimental *m/z*, isotopic ratio, and tandem MS spectra to lipid libraries, we confirmed 293 unique lipid species (**Supplementary file 5**). We found 107 lipids had a significant change in abundance between *bmm*[1] and *bmm*[rev] males ($p_{adj} < 0.05$): 85 species were upregulated in *bmm*[1] males and 22 lipid species were downregulated. Among differentially regulated species from different lipid classes, triglyceride had the largest residual above expected proportion ($p = 5.00 \times 10^{-4}$ by Pearson's Chi-squared test). This suggests triglyceride was the lipid class most affected by loss of *bmm* (**Figure 5B, C**).

In *bmm*[1] males, the majority of triglyceride species (55/97) were significantly higher in abundance compared with *bmm*[rev] control males. Because we observed a positive correlation between the fold increase in triglyceride abundance with both the number of double bonds ($p = 7.52 \times 10^{-8}$ by Kendall's rank correlation test; **Figure 5—figure supplement 1A**) and the number of carbons ($p = 2.77 \times 10^{-10}$ by Kendall's rank correlation test; **Figure 5—figure supplement 1B**), our data align well with *bmm/ATGL's* known role in regulating triglyceride levels (**Grönke et al., 2005**; **Haemmerle et al., 2006**; **Zimmermann et al., 2004**) and its substrate preference of long-chain polyunsaturated fatty acids (**Eichmann et al., 2012**). While we also detected changes in species such as fatty acids, acylcarnitine, and membrane lipids (**Figure 5—figure supplement 1C–H**), in line with recent *Drosophila* lipidomic data (**Nazario-Yepiz et al., 2021**; **Giedt et al., 2021**), the striking accumulation of triglyceride in *bmm*[1] males suggested that excess testis triglyceride in *bmm*[1] males may contribute to their spermatogenic

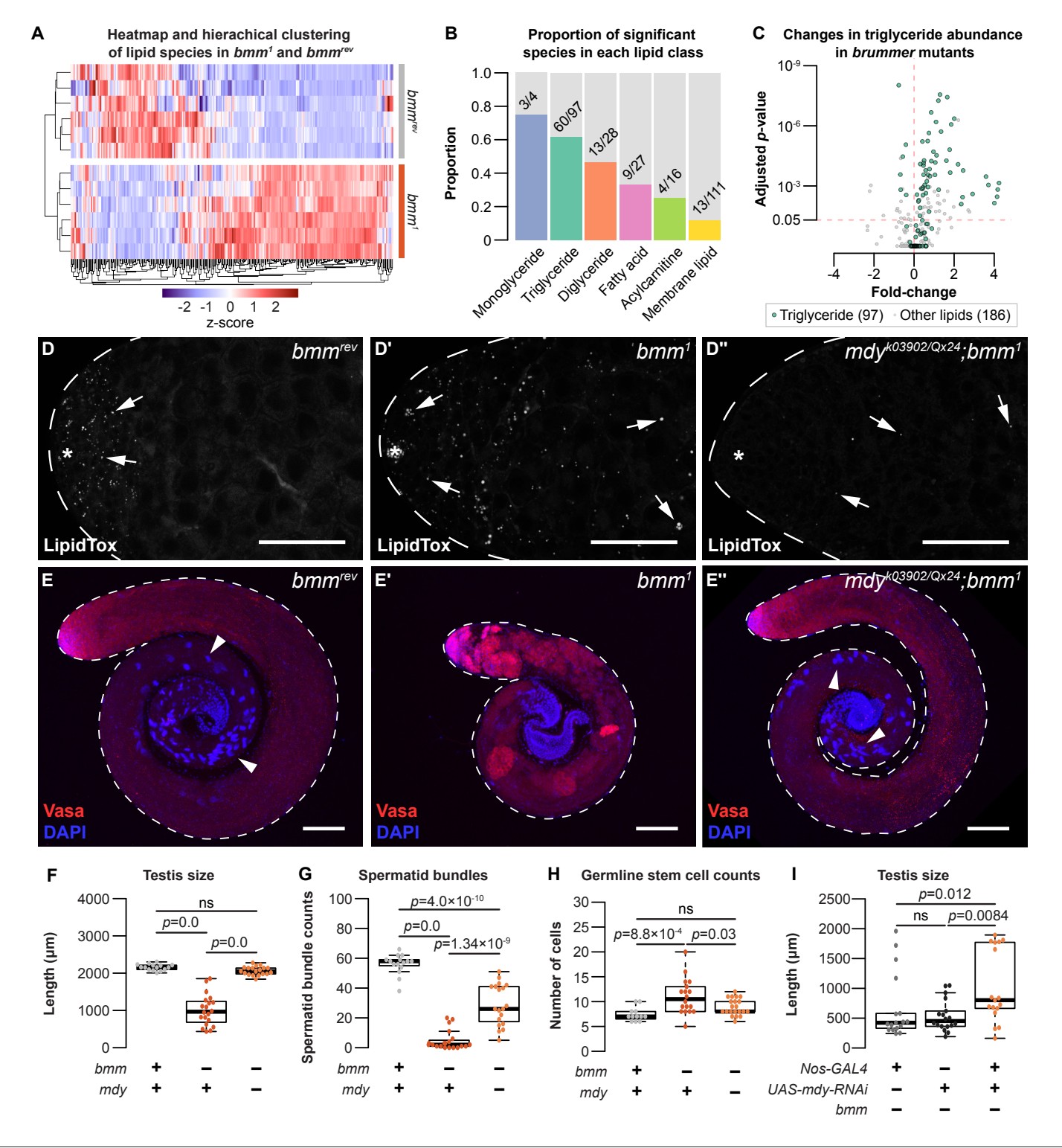

**Figure 5.** Loss of *bmm* disrupts triglyceride homeostasis and leads to spermatogenic defects. (**A**) Hierarchical clustering of lipid species detected in *bmm*^rev^ and *bmm*^1^ animals. (**B**) Histograms showing the proportion of significant species in each lipid class with different levels between *bmm*^1^ and *bmm*^rev^. Numbers on histograms indicate the number of species with differences in abundance. (**C**) Volcano plot showing fold change in abundance of triglyceride (green; 97 species) and non-triglyceride lipids (gray; 186 species) in our dataset. (**D**) Arrows indicate testis lipid droplets (LD) stained with LipidTox Red in *bmm*^rev^ (**D**), *bmm*^1^ (**D'**), or *mdy*^QX25/k03902^; *bmm*^1^ (**D''**) animals. (**E**) Whole testes isolated from *bmm*^rev^ (**E**), *bmm*^1^ (**E'**), or *mdy*^QX25/k03902^; *bmm*^1^ (**E''**) animals stained with anti-Vasa antibody (red) and DAPI (blue). Arrowheads indicate spermatid bundles. Scale bars = 100 μm. (**F**) Testis size in

*Figure 5 continued on next page*

*Figure 5 continued*

*bmm^rev*, *bmm^1*, and *mdy^QX25/k03902*;*bmm^1* animals. Spermatid bundles (**G**) and number of germline stem cells (**H**) in *bmm^rev*, *bmm^1*, and *mdy^QX25/k03902*;*bmm^1* animals. (**I**) Testis size in animals with germline-specific *mdy* knockdown (*nos-GAL4>mdy RNAi; bmm^1*) compared with controls (*nos-GAL4>+; bmm^1* and *+>mdy RNAi; bmm^1*). Error bars indicate standard error of the mean (SEM). See also *Figure 5—figure supplement 1*.

The online version of this article includes the following figure supplement(s) for figure 5:

**Figure supplement 1.** Lipidomic analysis of animals lacking *bmm*.

defects. To test this, we examined spermatogenesis in *bmm^1* males carrying loss-of-function mutations in *midway* (*mdy*). *mdy* is the *Drosophila* homolog of *diacylglycerol O-acyltransferase 1* (*DGAT1*), and whole-body loss of *mdy* reduces whole-body triglyceride levels (*Beller et al., 2010*; *Buszczak et al., 2002*; *Martínez et al., 2020*). Importantly, testes isolated from males with global loss of both *bmm* and *mdy* (*mdy^QX25/k03902*;*bmm^1*) had fewer LD than testes dissected from *bmm^1* males (*Figure 5D*, *Figure 5—figure supplement 1I*; one-way ANOVA with Tukey multiple comparison test).

We found that testes isolated from *mdy^QX25/k03902*;*bmm^1* males were significantly larger and had more spermatid bundles than testes from *bmm^1* males (*Figure 5E–G*; one-way ANOVA with Tukey multiple comparison test). The elevated number of GSCs in *bmm^1* male testes was similarly rescued in *mdy^QX25/k03902*;*bmm^1* males (*Figure 5H*; one-way ANOVA with Tukey multiple comparison test). These data suggest that defective spermatogenesis in *bmm^1* males can be partly attributed to excess triglyceride accumulation. Notably, at least some of the effects of global *mdy* loss on *bmm^1* males can be attributed to the germline: RNAi-mediated knockdown of *mdy* in the germline of *bmm^1* males partially rescued the defects in testis size (*Figure 5I*; Kruskal–Wallis rank sum test with Dunn's multiple comparison test) and GSC variance (*Figure 5—figure supplement 1J*; $p = 4.5 \times 10^{-5}$ and $8.2 \times 10^{-3}$ by $F$-test from the GAL4- and UAS-only crosses, respectively). While future studies will need to test whether germline-specific loss of *mdy* also rescues spermatid number defects in *bmm^1* males, our data suggest *bmm*-mediated regulation of testis triglyceride plays a previously unrecognized role in regulating sperm development.

## Discussion

In this study, we used *Drosophila* to gain insight into how the neutral lipids, a major lipid class, contribute to sperm development. We describe the distribution of LD under normal physiological conditions in the *Drosophila* testis, and show that LD are present at the early stages of development in both somatic and germline cells. While many factors are known to regulate LD in nongonadal cell types, we reveal a cell-autonomous role for triglyceride lipase *bmm* in regulating testis LD during spermatogenesis. In particular, we identified a requirement for *bmm* in mediating the decrease in LD at the spermatogonia–spermatocyte transition. This regulation is important for sperm development, as our data indicate that loss of *bmm* causes a decrease in the number of differentiated cell types in the testis. This reduction in differentiated cell types may be attributed to a delay in differentiation, a block in differentiation, or to a loss of differentiated cells through cell death. Future studies will therefore be essential to resolve why *bmm* loss causes a reduction in differentiated cell types. Nevertheless, these defects in the number of differentiated cell types can be partially explained by the excess accumulation of triglyceride in flies lacking *bmm*, as global and cell-type-specific inhibition of triglyceride synthesis rescues multiple spermatogenic defects in *bmm* mutants. Together, our data reveal previously unrecognized roles for LD and triglyceride during spermatogenesis, and for *bmm* as an important regulator of testis LD and germline development under normal physiological conditions.

One key outcome of our study was increased knowledge of LD regulation and function in the testis. Despite rapidly expanding knowledge of LD in cell types such as adipocytes or skeletal muscle, less is known about how LD influence spermatogenesis under normal physiological conditions. In mammals, testis LD contain cholesterol and play a role in promoting steroidogenesis (*Freeman and Ascoli, 1982*). In flies, we show that LD are present in the testis, and that excess accumulation of these LD affects sperm development. In nongonadal cell types, triglycerides provide a rich source of fatty acids for cellular ATP production, lipid building blocks to support membrane homeostasis and growth, and metabolites that can act as signaling molecules (*Walther and Farese, 2012*). Because ATP production, lipid precursors, and lipid signaling all play roles in supporting normal sperm development (*Walther et al., 2017*), future studies will need to determine how each of these processes is affected when

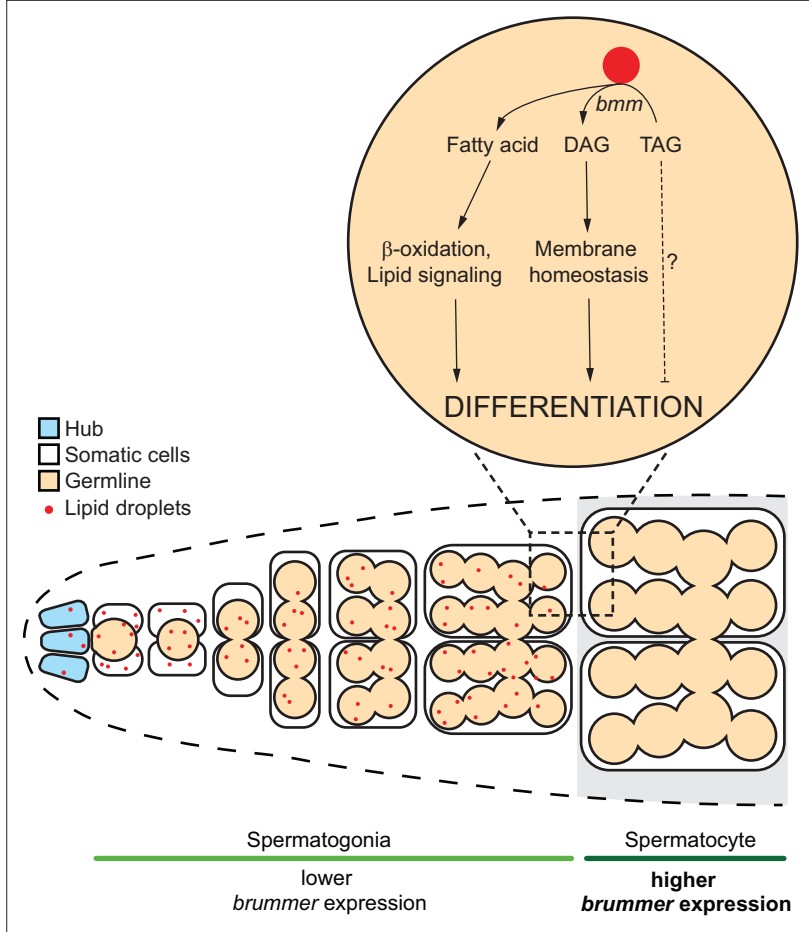

**Figure 6.** Model of *bmm*-mediated lipid droplet regulation in the *Drosophila* testis. Schematic representation summarizing *bmm*-mediated lipid droplet regulation in the testis during development.

excess triglyceride accumulates in testis LD (*Figure 6*). It will also be important to determine whether it is the loss of metabolites produced by *bmm*'s enzymatic action, or an increase in triglycerides, that leads to the reduction in differentiated cell types during spermatogenesis. Together, these experiments will provide critical insight into how triglyceride stored within testis LD contributes to overall cellular lipid metabolism during spermatogenesis. Because of the parallel spermatogenic defects we observed in *bmm* mutants and *ATGL*-deficient mice, we expect these mechanisms will also operate in other species.

A more comprehensive understanding of neutral lipid metabolism during sperm development will also emerge from studies on the upstream signaling networks that regulate testis LD and triglyceride. Given that we show an important and cell-autonomous role for *bmm* in regulating testis LD and triglyceride, future studies will need to identify factors that regulate *bmm* in the testis. Based on public single-cell RNAseq data and the *bmm-GFP* reporter strain, our data suggest *bmm* mRNA levels are differentially regulated between early and later stages of sperm development. Candidates for mediating this regulation include the insulin/insulin-like growth factor signaling pathway (IIS), target of rapamycin (TOR) pathway, and nuclear factor κB/Relish pathway (NFκB), as all of these pathways influence *bmm* mRNA levels in nongonadal cell types (*Birse et al., 2010*; *Molaei et al., 2019*; *Alic et al., 2011*; *Jünger et al., 2003*; *Zinke et al., 2002*; *Puig and Tjian, 2005*; *Kang et al., 2017*). Beyond mRNA levels, Bmm protein levels and post-translational modifications may also be differentially regulated during spermatogenesis. For example, studies show that the proteins encoded by *bmm* homologs in other animals are regulated by phosphorylation (*Bartz et al., 2007*) mediated by kinases such as adenosine monophosphate-activated protein kinase (AMPK) and protein kinase A (PKA) (*Pagnon et al., 2012*; *Narbonne and Roy, 2009*; *Ahmadian et al., 2011*). Importantly, many

of these pathways, including IIS, TOR, AMPK, NFκB, and possibly PKA influence *Drosophila* sperm development (*Amoyel et al., 2014*; *Hof-Michel et al., 2020*; *Couderc et al., 2017*; *Steinhauer et al., 2019*). Identifying the signaling networks that influence *bmm* regulation during sperm development will therefore lead to a deeper understanding of how testis LD and triglyceride are coordinated with physiological factors to promote normal spermatogenesis. Because pathways such as IIS and AMPK, and others, regulate sperm development in other species (*Tartarin et al., 2012*; *Martin-Hidalgo et al., 2018*; *Pitetti et al., 2013*), these insights may reveal conserved mechanisms that govern the regulation of cellular neutral lipid metabolism during sperm development.

# Materials and methods

**Key resources table**

| Reagent type (species) or resource | Designation | Source or reference | Identifiers | Additional information |
|---|---|---|---|---|
| Antibody | Anti-Vasa (rabbit, polyclonal) | Gift from Dr. R. Lehman, MIT | | IF (1:200) |
| Antibody | Anti-Eya (mouse monoclonal) | Developmental Studies Hybridoma Bank (DSHB) | eya10H6 | IF (1:50) |
| Antibody | Anti-zfh1 (mouse polyclonal) | Gift from Dr. J. Skeath, WUSTL | | IF (1:1000) |
| Antibody | Anti-boule (rabbit polyclonal) | Gift from Dr. S. Wasserman, UCSD | | IF (1:1000) |
| Antibody | Anti-phospho-histone H3 (mouse monoclonal) | Millipore Sigma | 05-1354 | IF (1:1000) |
| Strain, strain background | *w[1118]* | Bloomington *Drosophila* stock center | 3605 | 3605 |
| Strain, strain background | *CantonS* | Bloomington *Drosophila* stock center | 64349 | 64349 |
| Strain, strain background | *OregonR* | Bloomington *Drosophila* stock center | 25211 | 25211 |
| Strain, strain background | *bmm[1]* | Gift from Dr. R. Kühnlein; *Grönke et al., 2005* | | |
| Strain, strain background | *bmm[rev]* | Gift from Dr. R. Kühnlein; *Grönke et al., 2005* | | |
| Strain, strain background (*Drosophila melanogaster*) | *mdy[Qx25], cn[1], bw[1]/CyO, I(2)DTS513[1]* | Bloomington *Drosophila* stock center | 5095 | 5095 |
| Strain, strain background (*Drosophila melanogaster*) | *y[1],w[67c23];P{lacW}Cse1[k03802],mdy[k03902]/CyO* | Bloomington *Drosophila* stock center | 10536 | 10536 |
| Strain, strain background (*Drosophila melanogaster*) | *w[1118];P{GD1749}v6367 (UAS-mdy-RNAi)* | Vienna *Drosophila* resource center | 6367 | 6367 |
| Strain, strain background (*Drosophila melanogaster*) | *nos-GAL4::VP16* | Bloomington *Drosophila* stock center | 7303 | 7303 |
| Strain, strain background (*Drosophila melanogaster*) | *Tj-GAL4* | Gift from Dr. D. Godt, University of Toronto | | |

*Continued on next page*

*Continued*

| Reagent type (species) or resource | Designation | Source or reference | Identifiers | Additional information |
|---|---|---|---|---|
| Strain, strain background (*Drosophila melanogaster*) | c587-GAL4 | Bloomington *Drosophila* stock center | 67747 | 67747 |
| Strain, strain background (*Drosophila melanogaster*) | Bam-GFP | **Chen and McKearin, 2003** | | |
| Strain, strain background (*Drosophila melanogaster*) | bmm-GFP | Gift from Dr. K. Kamei; **Men et al., 2016** | | |
| Strain, strain background (*Drosophila melanogaster*) | GFP-LD | Gift from Dr. M. Welte; **Yu et al., 2011** | | |
| Strain, strain background (*Drosophila melanogaster*) | P{neoFRT}82B, bmm[1] | This study | | Flies available from E. Rideout, made as in 'Fly Husbandry' |
| Strain, strain background (*Drosophila melanogaster*) | P{neoFRT}82B, bmm[rev] | This study | | Flies available from E. Rideout, made as in 'Fly Husbandry' |
| Strain, strain background (*Drosophila melanogaster*) | bam-GFP, bmm[1] | This study | | Flies available from E. Rideout, made as in 'Fly Husbandry' |
| Strain, strain background (*Drosophila melanogaster*) | bam-GFP, bmm[rev] | This study | | Flies available from E. Rideout, made as in 'Fly Husbandry' |
| Software, algorithm | Fiji | https://imagej.net/software/fiji/ | | |
| Software, algorithm | R | https://cran.r-project.org | | |

## Materials and resource availability

*Drosophila* strains and their source are listed in the Key Resources table. Further information and requests for resources and reagents should be directed to, and will be fulfilled by, lead contact Dr. Elizabeth J. Rideout (elizabeth.rideout@ubc.ca).

## Fly husbandry

Fly stocks were maintained at room temperature in 12:12 hr light:dark cycle. Unless otherwise indicated, all flies were raised at 25°C with a density of 50 larvae per 10 ml fly media. Because this project examines sperm development, we used male flies in all experiments. Fly media contained 20.5 g sucrose (SU10, Snow Cap), 70.9 g Dextrose (SUG8, Snow Cap), 48.5 g cornmeal (AO18006, Snow Cap), 30.3 g baker's yeast (NB10, Snow Cap), 4.55 g agar (DR-820-25 F, SciMart), 0.5 g calcium chloride dihydrate (CCL302.1, BioShop Canada), 0.5 g magnesium sulfate heptahydrate (MAG511.1, BioShop Canada), 4.9 ml propionic acids (P1386, Sigma-Aldrich), and 488 µl phosphoric acid (P5811, Sigma-Aldrich) per 1 l of media. For diets with medium- or long-chain triglyceride, 4 g of coconut oil (medium-chain triglyceride) or olive oil (long-chain triglyceride) was added per 100 ml of media described above prior to cooling. Males were collected and dissected within 24 hr of eclosion unless otherwise indicated. Fixations were performed at room temperature with 4% paraformaldehyde (CA11021-168, VWR) in phosphate-buffered saline (PBS) for 20 min on a rotating platform followed by washing in PBS twice

before staining. Fly strains used in our study are listed in a Key Resources table, and fly strains prepared in this study were made using standard *Drosophila* genetic crossing techniques.

## Testis cell stage classification and measurements

Cells at an early stage of development (stem cells and early-stage somatic and germline cells) were located in the apical region of the testis, and were identified by their small and dense nuclei (**White-Cooper, 2004**). GSCs were defined as Vasa-positive cells in direct contact with the hub; proliferating GSCs were identified as Vasa-positive cells in direct contact with the hub that were also phospho-H3 positive. Cells in the testis region occupied by primary spermatocytes were identified by their large cell size and decondensed chromosome staining occupying three nuclear domains (**White-Cooper, 2004**). Spermatid bundles were identified by their condensed and needle-shaped nuclei, which roughly corresponds to nuclei with protamine-based chromatin (**Fabian and Brill, 2012**). The hub was identified as the FasIII-positive area of the testis. Hub size was estimated by measuring the FasIII-positive area in a Z-projected image of the hub in each testis. Z-projections were made using the 'sum slices' function in Fiji. Testis size was measured by quantifying the length of a line drawn down the middle of a testis image; starting from the apical tip of the testis and ending where the testis meets the seminal vesicle.

## FLP-FRT clone induction

Adult males were collected at 3–5 days post-eclosion and heat shocked three times at 37°C for 30 min followed by a 10-min rest period at room temperature between heat shocks. After heat shock, the flies were incubated at room temperature until dissection.

## Immunohistochemistry

Fixed samples were rinsed three times with blocking solution containing 0.2% bovine serum albumin (A4503, Sigma-Aldrich), 0.3% Triton-X in PBS, then blocked for 1 hr on a rotating platform at room temperature. During the incubation, the blocking solution was changed every 15 min. After blocking, the sample was resuspended in blocking solution with the appropriate concentration of primary antibody (see Key Resources table), and incubated overnight at 4°C. Samples were rinsed three times with blocking solution after removing primary antibody, and blocked for 1 hr on a rotating platform in blocking solution. Secondary antibody was applied in blocking solution and left on the rotating platform at room temperature for 40 min. The sample was rinsed with blocking solution three more times, and washed four times for 15 min per wash in blocking solution. Testis samples were resuspended in Vectashield mounting media with DAPI (H-1200-10, Vector Laboratory) or SlowFade Diamond mounting media (S36972, Thermo Fisher Scientific) prior to mounting.

## Lipid droplet staining

Fixed testes were briefly permeabilized with 0.1% Triton-X in PBS for 5 min prior to applying phalloidin. For BODIPY (4,4-Difluoro-1,3,5,7,8-Pentamethyl-4-Bora-3a,4a-Diaza-*s*-Indacene) staining, samples were suspended in PBS containing 10 µg/ml DAPI (2879083-5 mg, PeproTech), 1:500 BODIPY 495/503 (Thermo Fisher Scientific D3922), and 1:1000 phalloidin iFluor647 (ab176759, Abcam) or 1:40 phalloidin TexasRed (T7471, Thermo Fisher Scientific). For staining with LipidTox Red, samples were suspended in PBS containing 10 µg/ml DAPI (2879083-5 mg, PeproTech), 1:200 LipidTox Red (H34476, Thermo Fisher Scientific), and 1:1000 phalloidin iFluor647 (ab176759, Abcam). For staining free sterols, samples were prepared as for BODIPY staining with 50 µg/ml filipin in place of BODIPY for 30 min. Samples were incubated on a rotating platform for 40 min at room temperature. After incubation, samples were washed twice with PBS, then resuspended in SlowFade Diamond mounting media (Thermo Fisher Scientific S36972) prior to mounting.

## Image acquisition and processing

All images were acquired on a Leica SP5 confocal microscope system with ×20 or ×40 objectives and quantified with Fiji image analysis software (**Schindelin et al., 2012**).

## *Drosophila* lipidomics

*Drosophila* extracts were prepared following the previously reported protocol (**Yu et al., 2020**). Briefly, 10 *Drosophila* males (~10 mg) were weighed, 300 µl of ice-cold methanol/water mixture (9:1, vol:vol)

was added to these males, and the samples were homogenized with glass beads using a bead beater (mini-beadbeater-16, BioSpec, Bartlesville, OK, USA). Sample weight was used for sample normalization. Fly lysate was kept at −20°C for 4 hr for protein precipitation. Then, 900 µl of methyl tert-butyl ether was added and the solution was shaken for 5 min to extract lipids. To induce phase separation 285 µl of water was added, followed by centrifugation. The upper layer was separated, dried, and reconstituted in isopropanol/acetonitrile (1:1, vol:vol) for liquid chromatography (LC)–MS analysis. The volume of reconstitution solution was proportional to sample weight for normalization. Quality control (QC) samples were prepared by pooling 20 µl aliquot from each sample. The method blank sample was prepared using an identical workflow but without adding *Drosophila*.

*Drosophila* extracts were analyzed on an UHR-QqTOF (Ultra-High Resolution Qq-Time-Of-Flight) mass spectrometry Impact II (Bruker Daltonics, Bremen, Germany) interfaced with an Agilent 1290 Infinity II LC Systems (Agilent Technologies, Santa Clara, CA, USA). LC separation was performed using a Waters reversed-phase (RP) UPLC Acquity BEH C18 Column (1.7 µm, 1.0 mm × 100 mm, 130 Å) (Milford, MA, USA) maintained at 30°C. For positive ion mode, the mobile phase A was 60% acetonitrile in water and the mobile phase B was 90% isopropanol in acetonitrile, both containing 5 mM ammonium formate (pH = 4.8, adjusted by formic acid). For negative ion mode, the mobile phase A was 60% acetonitrile in water and the mobile phase B was 90% isopropanol in acetonitrile, both containing 5 mM $NH_4FA$ (pH = 9.8, adjusted by ammonium hydroxide). The LC gradient for positive and negative ion modes was set as follows: 0 min, 5% B; 8 min, 40% B; 14 min, 70% B; 20 min, 95% B; 23 min, 95% B; 24 min, 5% B; 33 min, 5% B. The flow rate was 0.1 ml/min. The injection volume was optimized to 2 µl in positive mode and 5 µl in negative mode using QC sample. The electrospray ionization (ESI) source conditions were set as follows: dry gas temperature, 220°C; dry gas flow, 7 l/min; nebulizer gas pressure, 1.6 bar; capillary voltage, 4500 V for positive mode, and 3000 V for negative mode. The MS1 analysis was conducted using following parameters: mass range, 70–1000 *m/z*; spectrum type: centroid, calculated using maximum intensity; absolute intensity threshold: 250. Data-dependent MS/MS analysis parameters: collision energy: 16–30 eV; cycle time, 3 s; spectra rate: 4 Hz when intensity $<10^4$ and 12 Hz when intensity $>10^5$, linearly increased from $10^4$ to $10^5$. External calibration was applied using sodium formate to ensure the *m/z* accuracy before sample analysis.

The raw LC–MS data were processed using MS-DIAL (ver. 4.38) (*Tsugawa et al., 2015*). The detailed MS-DIAL parameters are: MS1 tolerance, 0.01 Da; MS/MS tolerance, 0.05; mass slice width, 0.05 Da; smoothing method, linear weighted moving average; smoothing level, 3 scans; minimum peak width, 5 scans. Lipid features with high quantitative confidence were selected by the following criteria: retention time was within the gradient elution time (<23 min); average intensity in QC samples is larger than fivefold of the intensity in method blank sample. Lipid identification was performed by matching experimental precursor *m/z*, isotopic ratio and MS/MS spectrum against the LipidBlast libraries embedded in MS-DIAL. To improve the quantification accuracy, the measured MS signal intensities were corrected using serial diluted QC samples following the reported workflow (*Yu and Huan, 2021*).

## Quantification and statistical analysis

All microscopy images were quantified using Fiji software (*Schindelin et al., 2012*). For lipid droplet counts, a single optical slice through the middle of the testis containing the hub was used with the exception of FLP-FRT experiment where all LD within a GFP-negative cyst were counted (*Figure 2I*). All statistical analyses were done using R (obtained from https://cran.r-project.org). With exception of data concerning spatial distribution, and lipidomic data, Shapiro–Wilk test (via *shapiro.test* in base R) was used to assess normality of distribution prior to testing for significance. Kruskal–Wallis rank sum test (from the R package *coin*) and Dunn's test (from the R package *dunn.test*) were used in place of Welch two-sample *t*-test and Tukey's multiple comparison test when the assumption of normality was not met. For testing differences in variance between two populations, *F*-test (via *var.test* in base R) was used. For testing differences in spatial distribution, two-sample Kolmogorov–Smirnov test (via *ks.test* in base R) was used. All p-values are indicated in figures; extremely small p-values are listed as p < $2.2 \times 10^{-16}$.

## Acknowledgements

We thank Dr. Ronald Kühnlein for bmm[1] and bmm[rev] lines *Grönke et al., 2005*, Dr. Michael Welte for UAS-GFP-LD *Yu et al., 2011*, and Dr. Kaeko Kamei for bmm-GFP *Men et al., 2016*. We used stocks from the Bloomington *Drosophila* Stock Center (NIH P40OD018537) and Vienna *Drosophila* Resource Center (VDRC). We acknowledge critical resources and information provided by FlyBase *Thurmond et al., 2019* (supported by the National Human Genome Research Institute at the U.S. National Institutes of Health (U41 HG000739) and the British Medical Research Council (MR/N030117/1)). This work was supported by the Life Sciences Institutes Imaging Core, supported by the UBC GREx Biological Resilience Initiative. Funding for this study was provided by grants to EJR from the Canadian Institutes for Health Research (PJT-153072), Michael Smith Foundation for Health Research (16876), and the Canadian Foundation for Innovation (JELF-34879). GT was supported by a grant from the Natural Sciences and Engineering Research Council (NSERC; 2018-04648), TH/HY/CW were supported by NSERC (2020-04895), MA was supported by the Jacob's foundation. We would like to acknowledge that our research takes place on the traditional, ancestral, and unceded territory of the Musqueam people; a privilege for which we are grateful.

## Additional information

### Funding

| Funder | Grant reference number | Author |
| --- | --- | --- |
| Canadian Institutes of Health Research | PJT-153072 | Elizabeth Rideout |
| Natural Sciences and Engineering Research Council of Canada | 2018-04648 | Yanina-Yasmin Pesch Guy Tanentzapf |
| Natural Sciences and Engineering Research Council of Canada | 2020-04895 | Huaxu Yu Chenjingyi Wang Tao Huan |
| Michael Smith Health Research BC | 16876 | Elizabeth Rideout |
| Canada Foundation for Innovation | JELF-34879 | Elizabeth Rideout |

The funders had no role in study design, data collection, and interpretation, or the decision to submit the work for publication.

### Author contributions

Charlotte F Chao, Conceptualization, Formal analysis, Validation, Investigation, Visualization, Methodology, Writing – original draft, Writing – review and editing; Yanina-Yasmin Pesch, Formal analysis, Investigation, Visualization, Methodology, Writing – review and editing; Huaxu Yu, Chenjingyi Wang, Investigation, Methodology; Maria J Aristizabal, Methodology, Writing – review and editing; Tao Huan, Supervision, Funding acquisition, Investigation, Methodology, Writing – review and editing; Guy Tanentzapf, Supervision, Funding acquisition, Writing – review and editing; Elizabeth Rideout, Conceptualization, Supervision, Funding acquisition, Investigation, Methodology, Writing – original draft, Project administration, Writing – review and editing

### Author ORCIDs

Maria J Aristizabal ⓘ http://orcid.org/0000-0002-4491-6147
Elizabeth Rideout ⓘ http://orcid.org/0000-0003-0012-2828

Reviewer #1 (Public review): https://doi.org/10.7554/eLife.87523.4.sa1
Reviewer #2 (Public review): https://doi.org/10.7554/eLife.87523.4.sa2
Author response https://doi.org/10.7554/eLife.87523.4.sa3

## Additional files

### Supplementary files
- Supplementary file 1. Raw data and statistical outputs from *Figure 1*.
- Supplementary file 2. Raw data and statistical outputs from *Figures 2 and 3*.
- Supplementary file 3. Raw data and statistical outputs from *Figure 4*.
- Supplementary file 4. Raw data and statistical outputs from *Figure 5*.
- Supplementary file 5. Identified lipid species from untargeted lipidomic analysis.
- MDAR checklist

### Data availability
All data generated or analyzed during this study are included in the manuscript and supporting files.

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
