## [Editor Report · eLife assessment]

This **important** study identifies a role for triglycerides and lipid droplets in spermatogenesis, with data supporting relevance of this finding across phyla. The work shows with **convincing** data that a triglyceride lipase is required cell-autonomously for germline differentiation into meiotic stages and haploid spermatids and that an increase in triglycerides is detrimental to spermatogenesis. This paper would be of interest to developmental and cell biologists working on gametogenesis.

---

## [Referee Report · Reviewer #1 (Public review)]

In this study, the authors investigate the role of triglycerides in spermatogenesis. This work is based on their previous study (PMID: 31961851) on triglyceride sex differences in which they showed that somatic testicular cells play a role in whole body triglyceride homeostasis. In the current study, they show that lipid droplets (LDs) are significantly higher in the stem and progenitor cell (pre-meiotic) zone of the adult testis than in the meiotic spermatocyte stages. The distribution of LDs anti-correlates with the expression of the triglyceride lipase Brummer (Bmm), which has higher expression in spermatocytes than early germline stages. Analysis of a bmm mutant (bmm[1]) - a P-element insertion that is likely a hypomorphic - and its revertant (bmm[rev]) as a control shows that bmm acts autonomously in the germline to regulate LDs. In particular, the number of LDs is significantly higher in spermatocytes from bmm[1] mutants than from bmm[rev] controls. Testes from males with global loss of bmm (bmm[1]) are shorter than controls and have fewer differentiated spermatids. The zone of bam expression, typically close to the niche/hub in WT, is now many cell diameters away from the hub in bmm[1] mutants. There is an increase in the number of GSCs in bmm[1] homozygotes, but this phenotype is probably due to the enlarged hub. However, clonal analyses of GSCs lacking bmm indicate that a greater percentage of the GSC pool is composed of bmm[1]-mutant clones than of bmm[rev]-clones. This suggests that loss of bmm could impart a competitive advantage to GSCs, but this is not explored in greater detail. Despite the increase in number of GSCs that are bmm[1]-mutant clones, there is a significant reduction in the number of bmm[1]-mutant spermatocyte and post-meiotic clones. This suggests that fewer bmm[1]-mutant germ cells differentiate than controls. To gain insights into triglyceride homeostasis in the absence of bmm, they perform mass spec-based lipidomic profiling. Analyses of these data support their model that triglycerides are the class of lipid most affected by loss of bmm, supporting their model that excess triglycerides are the cause of spermatogenetic defects in bmm[1]. Consistent with their model, a double mutant of bmm[1] and a diacylglycerol O-acyltransferase 1 called midway (mdy) reverts the bmm-mutant germline phenotypes.

There are numerous strengths of this paper. First, the authors report rigorous measurements and statistical analyses throughout the study. Second, the authors utilize robust genetic analyses with loss-of-function mutants and lineage-specific knockdown. Third, they demonstrate the appropriate use of controls and markers. Fourth, they show rigorous lipidomic profiling. Lastly, their conclusions are appropriate for the results. In other words, they don't over-state the results. Overall, the rigorously quantified results support the major aim that appropriate regulation of triglycerides are needed in a germline cell-autonomous manner for spermatogenesis.

This paper should have a positive impact on the field. First and foremost, there is limited knowledge about the role of lipid metabolism in spermatogenesis. The lipidomic data will be useful to researchers in the field who study various lipid species. Going forward, it will be very interesting to determine what triglycerides regulate in germline biology. In other words, what functions/pathways/processes in germ cells are negatively impacted by elevated triglycerides. And as the authors point out in the discussion, it will be important to determine what regulates bmm expression such that bmm is higher in later stages of germline differentiation.

---

## [Referee Report · Reviewer #2 (Public review)]

Summary:

Here, the authors show that neutral lipids play a role in spermatogenesis. Neutral lipids are components of lipid droplets, which are known to maintain lipid homeostasis, and to be involved in non-gonadal differentiation, survival, and energy. Lipid droplets are present in the testis in mice and *Drosophila*, but not much is known about the role of lipid droplets during spermatogenesis. The authors show that lipid droplets are present in early differentiating germ cells, and absent in spermatocytes. They further show a cell autonomous role for the lipase brummer in regulating lipid droplets and, in turn, spermatogenesis in the *Drosophila* testis. The data presented show that a relationship between lipid metabolism and spermatogenesis is congruous in mammals and flies, supporting *Drosophila* spermatogenesis as an effective model to uncover the role lipid droplets play in the testis.

Strengths and weaknesses:

The authors do a commendably thorough characterization of where lipid droplets are detected in normal testes: located in young somatic cells, and early differentiating germ cells. They use multiple control backgrounds in their analysis, including w[1118], Canton S, and Oregon R, which adds rigor to their interpretations. The authors employ markers that identify which lipid droplets are in somatic cells, and which are in germ cells. The authors use these markers to present measured distances of somatic and germ cell-derived lipid droplets from the hub. Because they can also measure the distance of somatic and germ cells with age-specific markers from the hub, these results allow the authors to correlate position of lipid droplets with the age of cells in which they are present. This analysis is clearly shown and well quantified.

The quantification of lipid droplet distance from the hub is applied well in comparing brummer mutant testes to wild type controls. The authors measure the number of lipid droplets of specific diameters, and the spatial distribution of lipid droplets as a function of distance from the hub. These measurements quantitatively support their findings that lipid droplets are present in an expanded population of cells further from the hub in brummer mutants. The authors further quantify lipid droplets in germline clones of specified ages; the quantitative analysis here is displayed clearly and supports a cell autonomous role for brummer in regulating lipid droplets in spermatocytes.

Data examining testis size and number of spermatids in brummer mutants clearly indicates the importance of regulating lipid droplets to spermatogenesis. The authors show beautiful images supported by rigorous quantification supporting their findings that brummer mutants have both smaller testes with fewer spermatids at both 29 and 25C. There is also significant data supporting defects in testis size, but not spermatid number, in 14-day-old brummer mutant animals compared to controls. Their analysis clearly shows an expanded region beyond the testis apex that includes younger germ cells, supporting a role for lipid droplets influencing germ cell differentiation during spermatogenesis.

The authors present a series of data exploring a cell autonomous role for brummer in the germline, including clonal analysis and tissue specific manipulations. The clonal data indicating increased lipid droplets in spermatocyte clones, and a higher proportion of brummer mutant GSCs at the hub are convincing and supported by quantitation. The authors also show a tissue specific rescue of the brummer testis size phenotype by knocking down mdy specifically in germ cells, which is also supported by statistically significant quantitation. The authors present data examining the number of spermatocyte and post-meiotic clones 14 days after clonal induction. Their finding is significant with a p-value of 0.0496, which they acknowledge is less robust than their other data reported in this study, and could be a result of a low sample size. They indicate that future studies might validate these results with additional samples.

The authors do a beautiful job of validating where they detect brummer-GFP by presenting their own pseudotime analysis of publicly available single cell RNA sequencing data. Their data is presented very clearly, and supports expression of brummer in older somatic and germline cells of the age when lipid droplets are normally not detected. The authors also present a thorough lipidomic analysis of animals lacking brummer to identify triglycerides as an important lipid droplet component regulating spermatogenesis.

Impact:

The authors present data supporting the broad significance of their findings across phyla. This data represents a key strength of this manuscript. The authors show that loss of a conserved triglyceride lipase impacts testis development and spermatogenesis, and that these impacts can be rescued by supplementing diet with medium-chain triglycerides. The authors point out that these findings represent a biological similarity between *Drosophila* and mice, supporting the relevance of the *Drosophila* testis as a model for understanding the role of lipid droplets in spermatogenesis. The connection buttresses the relevance of these findings and this model to a broad scientific community.

---

## [Author Response]

The following is the authors’ response to the previous reviews.

**PUBLIC REVIEWS**

**Reviewer #1 (Public Review):**
In this study, the authors investigate the role of triglycerides in spermatogenesis. This work is based on their previous study (PMID: 31961851) on triglyceride sex differences in which they showed that somatic testicular cells play a role in whole body triglyceride homeostasis. In the current study, they show that lipid droplets (LDs) are significantly higher in the stem and progenitor cell (pre-meiotic) zone of the adult testis than in the meiotic spermatocyte stages. The distribution of LDs anti-correlates with the expression of the triglyceride lipase Brummer (Bmm), which has higher expression in spermatocytes than early germline stages. Analysis of a bmm mutant (bmm[1]) - a P-element insertion that is likely a hypomorphic - and its revertant (bmm[rev]) as a control shows that bmm acts autonomously in the germline to regulate LDs. In particular, the number of LDs is significantly higher in spermatocytes from bmm[1] mutants than from bmm[rev] controls. Testes from males with global loss of bmm (bmm[1]) are shorter than controls and have fewer differentiated spermatids. The zone of bam expression, typically close to the niche/hub in WT, is now many cell diameters away from the hub in bmm[1] mutants. There is an increase in the number of GSCs in bmm[1] homozygotes, but this phenotype is probably due to the enlarged hub. However, clonal analyses of GSCs lacking bmm indicate that a greater percentage of the GSC pool is composed of bmm[1]-mutant clones than of bmm[rev]-clones. This suggests that loss of bmm could impart a competitive advantage to GSCs, but this is not explored in greater detail. Despite the increase in number of GSCs that are bmm[1]-mutant clones, there is a significant reduction in the number of bmm[1]-mutant spermatocyte and post-meiotic clones. This suggests that fewer bmm[1]mutant germ cells differentiate than controls. To gain insights into triglyceride homeostasis in the absence of bmm, they perform mass spec-based lipidomic profiling. Analyses of these data support their model that triglycerides are the class of lipid most affected by loss of bmm, supporting their model that excess triglycerides are the cause of spermatogenetic defects in bmm[1]. Consistent with their model, a double mutant of bmm[1] and a diacylglycerol Oacyltransferase 1 called midway (mdy) reverts the bmm-mutant germline phenotypes.There are numerous strengths of this paper. First, the authors report rigorous measurements and statistical analyses throughout the study. Second, the authors utilize robust genetic analyses with loss-of-function mutants and lineage-specific knockdown. Third, they demonstrate the appropriate use of controls and markers. Fourth, they show rigorous lipidomic profiling. Lastly, their conclusions are appropriate for the results. In other words, they don't over-state the results. Overall, the rigorously quantified results support the major aim that appropriate regulation of triglycerides are needed in a germline cell-autonomous manner for spermatogenesis.This paper should have a positive impact on the field. First and foremost, there is limited knowledge about the role of lipid metabolism in spermatogenesis. The lipidomic data will be useful to researchers in the field who study various lipid species. Going forward, it will be very interesting to determine what triglycerides regulate in germline biology. In other words, what functions/pathways/processes in germ cells are negatively impacted by elevated triglycerides. And as the authors point out in the discussion, it will be important to determine what regulates bmm expression such that bmm is higher in later stages of germline differentiation.

We thank the Reviewer for their positive assessment of our revised manuscript!

**Reviewer #2 (Public Review):**
Summary:Here, the authors show that neutral lipids play a role in spermatogenesis. Neutral lipids are components of lipid droplets, which are known to maintain lipid homeostasis, and to be involved in non-gonadal differentiation, survival, and energy. Lipid droplets are present in the testis in mice and *Drosophila*, but not much is known about the role of lipid droplets during spermatogenesis. The authors show that lipid droplets are present in early differentiating germ cells, and absent in spermatocytes. They further show a cell autonomous role for the lipase brummer in regulating lipid droplets and, in turn, spermatogenesis in the *Drosophila* testis. The data presented show that a relationship between lipid metabolism and spermatogenesis is congruous in mammals and flies, supporting *Drosophila* spermatogenesis as an effective model to uncover the role lipid droplets play in the testis.Strengths and weaknesses:The authors do a commendably thorough characterization of where lipid droplets are detected in normal testes: located in young somatic cells, and early differentiating germ cells. They use multiple control backgrounds in their analysis, including w[1118], Canton S, and Oregon R, which adds rigor to their interpretations. The authors employ markers that identify which lipid droplets are in somatic cells, and which are in germ cells. The authors use these markers to present measured distances of somatic and germ cell-derived lipid droplets from the hub. Because they can also measure the distance of somatic and germ cells with age-specific markers from the hub, these results allow the authors to correlate position of lipid droplets with the age of cells in which they are present. This analysis is clearly shown and well quantified.The quantification of lipid droplet distance from the hub is applied well in comparing brummer mutant testes to wild type controls. The authors measure the number of lipid droplets of specific diameters, and the spatial distribution of lipid droplets as a function of distance from the hub. These measurements quantitatively support their findings that lipid droplets are present in an expanded population of cells further from the hub in brummer mutants. The authors further quantify lipid droplets in germline clones of specified ages; the quantitative analysis here is displayed clearly and supports a cell autonomous role for brummer in regulating lipid droplets in spermatocytes.Data examining testis size and number of spermatids in brummer mutants clearly indicates the importance of regulating lipid droplets to spermatogenesis. The authors show beautiful images supported by rigorous quantification supporting their findings that brummer mutants have both smaller testes with fewer spermatids at both 29 and 25C. There is also significant data supporting defects in testis size, but not spermatid number, in 14-day-old brummer mutant animals compared to controls. Their analysis clearly shows an expanded region beyond the testis apex that includes younger germ cells, supporting a role for lipid droplets influencing germ cell differentiation during spermatogenesis.The authors present a series of data exploring a cell autonomous role for brummer in the germline, including clonal analysis and tissue specific manipulations. The clonal data indicating increased lipid droplets in spermatocyte clones, and a higher proportion of brummer mutant GSCs at the hub are convincing and supported by quantitation. The authors also show a tissue specific rescue of the brummer testis size phenotype by knocking down mdy specifically in germ cells, which is also supported by statistically significant quantitation. The authors present data examining the number of spermatocyte and post-meiotic clones 14 days after clonal induction. Their finding is significant with a p-value of 0.0496, which they acknowledge is less robust than their other data reported in this study, and could be a result of a low sample size. They indicate that future studies might validate these results with additional samples.The authors do a beautiful job of validating where they detect brummer-GFP by presenting their own pseudotime analysis of publicly available single cell RNA sequencing data. Their data is presented very clearly, and supports expression of brummer in older somatic and germline cells of the age when lipid droplets are normally not detected. The authors also present a thorough lipidomic analysis of animals lacking brummer to identify triglycerides as an important lipid droplet component regulating spermatogenesis.Impact:The authors present data supporting the broad significance of their findings across phyla. This data represents a key strength of this manuscript. The authors show that loss of a conserved triglyceride lipase impacts testis development and spermatogenesis, and that these impacts can be rescued by supplementing diet with medium-chain triglycerides. The authors point out that these findings represent a biological similarity between *Drosophila* and mice, supporting the relevance of the *Drosophila* testis as a model for understanding the role of lipid droplets in spermatogenesis. The connection buttresses the relevance of these findings and this model to a broad scientific community.

We thank the Reviewer for their positive assessment of our revised paper!

**Reviewer #2 (Recommendations For The Authors):**
The authors addressed most of my recommendations in a way that is satisfactory to me. I would like a bit more information added to the methods section about how hub area was quantified. For example, did the authors measure area within a defined region in a single Z plane (perhaps the Z plane at the center of the hub, or the Z plane with the largest area)?Alternatively, did they authors measure area in a more 3 dimensional way, i.e. volume.Adding this information to the methods would satisfy all of my previous recommendations.

We thank the Reviewer for pointing out that this information was not clear in the revised manuscript. We changed the methods section to clarify our methods as follows:

“The hub was identified as the FasIII-positive area of the testis. Hub size was estimated by measuring the FasIII-positive area in a Z-projected image of the hub in each testis. Zprojections were made using the ‘sum slices’ function in Fiji.”